# Initial Trans-Arterial Chemo-Embolisation (TACE) Is Associated with Similar Survival Outcomes as Compared to Upfront Percutaneous Ablation Allowing for Follow-Up Treatment in Those with Single Hepatocellular Carcinoma (HCC) ≤ 3 cm: Results of a Real-World Propensity-Matched Multi-Centre Australian Cohort Study

**DOI:** 10.3390/cancers16173010

**Published:** 2024-08-29

**Authors:** Jonathan Abdelmalak, Simone I. Strasser, Natalie L. Ngu, Claude Dennis, Marie Sinclair, Avik Majumdar, Kate Collins, Katherine Bateman, Anouk Dev, Joshua H. Abasszade, Zina Valaydon, Daniel Saitta, Kathryn Gazelakis, Susan Byers, Jacinta Holmes, Alexander J. Thompson, Jessica Howell, Dhivya Pandiaraja, Steven Bollipo, Suresh Sharma, Merlyn Joseph, Rohit Sawhney, Amanda Nicoll, Nicholas Batt, Myo J. Tang, Stephen Riordan, Nicholas Hannah, James Haridy, Siddharth Sood, Eileen Lam, Elysia Greenhill, John Lubel, William Kemp, Ammar Majeed, John Zalcberg, Stuart K. Roberts

**Affiliations:** 1Department of Gastroenterology, Alfred Health, Melbourne, VIC 3004, Australia; j.abdelmalak@alfred.org.au (J.A.); tangmyojin@gmail.com (M.J.T.); j.lubel@alfred.org.au (J.L.); w.kemp@alfred.org.au (W.K.); a.majeed@alfred.org.au (A.M.); 2Department of Medicine, School of Translational Medicine, Monash University, Melbourne, VIC 3004, Australia; eileen.lam@monash.edu (E.L.); elysia.greenhill@monash.edu (E.G.); 3Department of Gastroenterology, Austin Hospital, Heidelberg, VIC 3084, Australia; marie.sinclair@austin.org.au (M.S.); avik.majumdar@austin.org.au (A.M.); kate.collins3@austin.org.au (K.C.); kat.bateman@austin.org.au (K.B.); 4AW Morrow Gastroenterology and Liver Centre, Royal Prince Alfred Hospital, Camperdown, NSW 2050, Australia; simone.strasser@health.nsw.gov.au (S.I.S.); natalielyngu@gmail.com (N.L.N.); claude.dennis@health.nsw.gov.au (C.D.); 5Department of Gastroenterology, Monash Health, Clayton, VIC 3168, Australia; anouk.dev@monash.edu (A.D.); joshua.abasszade@monashhealth.org (J.H.A.); 6Department of Gastroenterology, Western Health, Footscray, VIC 3011, Australia; zina.valaydon@wh.org.au (Z.V.); daniel.saitta@wh.org.au (D.S.); kathryn.gazelakis@wh.org.au (K.G.); susan.byers1@wh.org.au (S.B.); 7Department of Gastroenterology, St. Vincent’s Hospital Melbourne, Fitzroy, VIC 3065, Australia; jacinta.holmes@svha.org.au (J.H.); alexander.thompson@svha.org.au (A.J.T.); jessica.howell@svha.org.au (J.H.); dhivya.pandiaraja@svha.org.au (D.P.); 8Department of Medicine, St. Vincent’s Hospital Melbourne, University of Melbourne, Parkville, VIC 3052, Australia; 9Department of Gastroenterology, John Hunter Hospital, New Lambton Heights, NSW 2305, Australia; steven.bollipo@health.nsw.gov.au (S.B.); suresh.sharma@health.nsw.gov.au (S.S.); merlyn.joseph@health.nsw.gov.au (M.J.); 10Department of Gastroenterology, Eastern Health, Box Hill, VIC 3128, Australia; rohit.sawhney@easternhealth.org.au (R.S.); amanda.nicoll@easternhealth.org.au (A.N.); nicholas.batt2@austin.org.au (N.B.); 11Department of Medicine, Eastern Health Clinical School, Box Hill, VIC 3128, Australia; 12Department of Gastroenterology, Prince of Wales Hospital, Randwick, NSW 2031, Australia; stephen.riordan@health.nsw.gov.au; 13Department of Gastroenterology, Royal Melbourne Hospital, Parkville, VIC 3052, Australia; nicholas.hannah2@mh.org.au (N.H.); sood.s@unimelb.edu.au (S.S.); 14School of Public Health and Preventive Medicine, Monash University, Melbourne, VIC 3004, Australia; john.zalcberg@monash.edu; 15Department of Medical Oncology, Alfred Health, Melbourne, VIC 3004, Australia

**Keywords:** hepatocellular carcinoma, single, small, early, ablation, TACE

## Abstract

**Simple Summary:**

Single small primary liver cancers are curable with treatments such as surgical resection, ablation, and liver transplant; however, many patients initially receive trans-arterial chemo-embolisation (TACE), which is generally not considered a curative treatment in itself, and often go on to receive further follow-up treatments. Little is known regarding the outcomes of such patients compared to those who receive upfront ablation. Our real-world multi-centre study demonstrates that key survival outcomes are similar between those initially undergoing TACE and those receiving ablation after controlling for other key clinical variables and allowing for subsequent individualised treatment selection.

**Abstract:**

Percutaneous ablation is recommended in Barcelona Clinic Liver Cancer (BCLC) stage 0/A patients with HCC ≤3 cm as a curative treatment modality alongside surgical resection and liver transplantation. However, trans-arterial chemo-embolisation (TACE) is commonly used in the real-world as an initial treatment in patients with single small HCC in contrast to widely accepted clinical practice guidelines which typically describe TACE as a treatment for intermediate-stage HCC. We performed this real-world propensity-matched multi-centre cohort study in patients with single HCC ≤ 3 cm to assess for differences in survival outcomes between those undergoing initial TACE and those receiving upfront ablation. Patients with a new diagnosis of BCLC 0/A HCC with a single tumour ≤3 cm first diagnosed between 1 January 2016 and 31 December 2020 who received initial TACE or ablation were included in the study. A total of 348 patients were included in the study, with 147 patients receiving initial TACE and 201 patients undergoing upfront ablation. After propensity score matching using key covariates, 230 patients were available for analysis with 115 in each group. There were no significant differences in overall survival (log-rank test *p* = 0.652) or liver-related survival (log-rank test *p* = 0.495) over a median follow-up of 43 months. While rates of CR were superior after ablation compared to TACE as a first treatment (74% vs. 56%, *p* < 0.004), there was no significant difference in CR rates when allowing for further subsequent treatments (86% vs. 80% *p* = 0.219). In those who achieved CR, recurrence-free survival and local recurrence-free survival were similar (log rank test *p* = 0.355 and *p* = 0.390, respectively). Our study provides valuable real-world evidence that TACE when offered with appropriate follow-up treatment is a reasonable initial management strategy in very early/early-stage HCC, with similar survival outcomes as compared to those managed with upfront ablation. Further work is needed to better define the role for TACE in BCLC 0/A HCC.

## 1. Introduction

Hepatocellular carcinoma (HCC) is the third most common cause of cancer-related death, accounting for 830,000 deaths in 2020 [1], and is projected to continue to rise in incidence [2]. The Barcelona Clinic Liver Cancer (BCLC) staging system is widely used as a framework in considering the staging, management, and prognosis of HCC, by factoring in tumour burden, liver disease severity, and cancer-related performance status using the Eastern Cooperative Oncology Group (ECOG) scale. Patients with a single small HCC without extrahepatic disease, and in the presence of preserved liver function and performance status, are considered candidates for curative therapy and have the best prognosis compared to those with multinodular or otherwise more advanced disease. Those with tumour ≤2 cm are defined as BCLC 0, while those with single tumours > 2 cm are described as BCLC A.

In patients with small (≤3 cm) solitary HCC, resection and ablation are the leading treatment modalities recommended for curative intent [3]. However, trans-arterial chemo-embolisation (TACE), which has traditionally been described as a treatment for intermediate-stage or multinodular HCC rather than small solitary HCC, is widely used as an alternative modality where anatomical or patient-related morbidity considerations preclude resection and ablation, often with an intention of future curative-intent treatment such as ablation or transplantation. Indeed, we recently reported that in some centres, up to 60% of patients with solitary small HCC undergo TACE as their initial treatment [4]. Despite the widespread use of TACE in these patients, there is a paucity of published literature describing the outcomes associated with its use compared to more conventional treatments. Only a small number of studies [5,6,7,8] have reported on this question, with evidence of similar outcomes as compared to ablation; however, the generalisability of these findings requires further verification, as all of these studies were single-centre and none of them reported on subsequent follow-up treatment. In order to describe the real-world use and outcomes associated with a treatment strategy involving initial TACE for single small HCC, together with and without further follow-up treatment, and provide more robust evidence in comparing its impact on survival outcomes as with upfront ablation, we performed this multi-centre retrospective observational propensity-score-matched cohort study.

## 2. Materials and Methods

### 2.1. Participants

Our study included subjects with a diagnosis of HCC made between 1 January 2016 and 31 December 2020 across ten Australian tertiary liver cancer referral centres across Victoria and New South Wales, including two centres with an integrated liver transplant program. Inclusion criteria were as follows: adult aged >18 years of age; first diagnosis of HCC documented between 1 January 2016 and 31 December 2020 on the basis of imaging fulfilling LIRAD-5 criteria or histology confirming HCC; single lesion 3 cm or less, CP A or B, cancer-related performance status of ECOG 0, and absence of extrahepatic disease or vascular invasion; and received as first treatment either ablative therapy including microwave ablation (MWA), radiofrequency ablation (RFA) or percutaneous ethanol injection (PEI), or conventional or drug-eluting bead trans-arterial chemo-embolisation (cTACE or DEB TACE). Exclusion criteria were prior diagnosis or past history of HCC; more than one liver tumour; tumour size >3 cm; diagnosis of other solid organ malignancy other than non-melanotic skin cancer; and insufficient data available to sufficiently describe HCC stage. Ethics for the study was approved by Monash Health Human Research Ethics Committee (HREC) with approval for waiver of consent. All patient data were deidentified during data entry.

### 2.2. Study Design

Our multicentre retrospective cohort study involved retrospective data collection involving review of the medical record from the date of initial diagnosis of HCC to the end of follow-up (either death or last medical record entry available at time of data extraction). Demographic, clinical, biochemical, and imaging data were collected in pre-specified pro-forma. Treatment details including modalities and dates of initial and subsequent treatments, and outcome details, including dates and findings of response assessment imaging and mortality, were collected in a repeating data field. Modified RECIST criteria (mRECIST) [9] were applied at all ten sites as standard of care in the interpretation of imaging obtained during follow-up. ‘Complete response’ (CR) is defined as the disappearance of arterial enhancement within all target lesions. The minimum dataset is outlined in detail in Appendix B. All data were entered in a deidentified form into a centralised REDCap electronic data capture tool hosted by Monash University.

### 2.3. Endpoints

The primary endpoint in our study was overall survival (OS), which is defined as time from initial diagnosis of HCC to death. Secondary endpoints included (a) liver-related survival (LRS), which is defined as time from diagnosis to liver-related death (with non-liver death considered a censoring event); (b) all-cause mortality with liver transplant as competing event; and (c) liver-related mortality with liver transplant as a competing event. We also reported the rates of attainment of CR after initial and multiple lines of therapy. In those who required multiple lines of therapy before CR, we reported the sequence of treatments required to achieve CR. In all those who achieved CR, including those who achieved CR after multiple sequences of different therapies, we assessed for recurrence-free survival (RFS), which is defined as the time from first imaging documenting CR to either death or the date of imaging documenting HCC recurrence. We also reported local recurrence-free survival (LRFS), which is defined as the time from first imaging documenting CR to death or time of imaging considered to demonstrate recurrent tumour at the site of prior treatment as deemed by the documenting clinician. As all patients who underwent transplant either did so before achieving CR or after a recurrence, we did not need to perform RFS/LRFS analysis with competing risks regression using transplant as a competing event.

### 2.4. Statistical Analysis

SPSS 29.0 (SPSS, Inc., Chicago, IL, USA) and STATA 18 software (StataCorp. 2023, College Station, TX, USA) were used for data analysis. We performed binary logistic regression, using a forward-selection strategy, to determine the factors predicting treatment group allocation (upfront ablation or initial TACE). The results of the binary logistic regression are presented in Appendix A. The variables deemed clinically significant and therefore used to calculate the propensity score were as follows: age, sex, diabetes, smoking, HBV, alcohol, tumour size (cm), platelet count, CP score, and Charlson Comorbidity Index (CCI) [10]. Nearest-neighbour propensity score matching in a 1:1 ratio was performed using a match tolerance of 0.01, which was the highest value where all relevant clinical covariates were adequately comparable between groups.

We constructed a histogram of the distribution of propensity scores in each group before and after matching to verify that matching had been successful. We reported the statistical significance of differences between the two groups before and after propensity score matching was assessed using the chi-squared test for categorical variables, Mann–Whitney U test for non-parametric variables, and independent sample *t*-test for parametric variables.

Kaplan–Meier survival analysis with log-rank test was used to assess OS, LRS RFS, and LRFS in the ablation and TACE groups and the statistical significance of any differences observed. Survival rates at 1 and 3 years were extrapolated from Kaplan–Meier survival analysis, and the log-rank test with censorement at these pre-specified time points was utilised to assess for the statistical significance of survival differences up until then. Kaplan–Meier analysis was also performed in the original unmatched cohort to assess for unadjusted survival differences. To reduce the risk of competing-risk bias due to liver transplantation, which is known to significantly reduce risk of death and could be associated with preferential treatment with TACE over ablation, we performed competing-risks regression analysis, with liver transplantation as a competing risk for all-cause death and also liver-related death. Using the STATA 18 statistical software, we calculated adjusted hazard ratios with 95% confidence intervals and plotted cumulative incidence function curves.

The two-tailed *p* < 0.05 was considered statistically significant in all analyses used in the study.

## 3. Results

### 3.1. Patients

A total of 348 patients were included in our study, with 201 receiving upfront ablation and 147 undergoing initial TACE. Median follow-up time from diagnosis to death or censorement was 43 months (1308.5 days).

In the original unmatched cohort, there were evident systematic differences between those in the ablation and TACE groups. Patients who were selected for TACE had a greater median tumour size (2.1 cm vs. 1.8 cm, *p* < 0.001) and were numerically under-represented with BCLC-0 disease (42% vs. 52%, *p* = 0.059). Liver disease aetiology also appeared systematically different, with ablation patients more likely to have alcohol as a cause for their liver disease as compared to TACE patients. Liver disease severity was overall similar between the two groups (*p* = 0.078), although there was a numerically higher proportion of CPB 8/9 cirrhosis in those undergoing TACE (14% vs. 6%). Age, sex, diabetes, smoking, platelet count, and CCI were otherwise similar between the two groups. A total of 107 out of 147 (73%) of the initial TACE patients were managed at a liver transplant centre compared to 64 out of 201 (32%) upfront ablation patients (in the PSM cohort, 87 out of 115 (76%) versus 47 out of 115 (41%)). This stark difference reflects a systematic difference in treatment approach between liver transplant and non-transplant centres that we have previously described [4]. Because treatment strategy with initial TACE was directly linked with management at a liver transplant centre, the managing centre was not used in the propensity score matching process to avoid collinearity.

Propensity score matching produced a total of 115 matched pairs (total cohort of 230 patients). The propensity-score-matched (PSM) cohort was well-matched on all relevant clinical covariates, with no significant differences seen between the two groups. Patient characteristics before and after matching are presented in Table 1. Histogram of propensity scores in the two groups before and after matching are presented in Figure 1, with well-matched distributions seen after matching.

### 3.2. Sequential Treatment Prior to Complete Response

Table 2 describes in detail the sequence of treatments needed to achieve complete response (CR) in the ablation and TACE groups.

In the ablation group, 14% of patients never achieved CR, 75% achieved CR after the initial treatment, and 11% achieved CR after more than one treatment. Of the 21 ablation patients who needed more than one treatment to achieve CR, 9 patients underwent sequential ablation, 8 underwent a combination of sequential ablation and TACE, 2 ultimately underwent resection, 1 received SBRT, and another received selective internal radiation therapy (SIRT). No patients who initially received ablation received liver transplant as a salvage treatment.

In the TACE group, 20% of TACE patients never achieved CR, 53% achieved CR after the initial treatment, 15% achieved CR after with TACE in addition to follow-up subsequent ablation, 9% after multiple TACE, 2% after salvage liver transplant, and 1% after eventual resection. Of the 22 patients who achieved CR after TACE with follow-up ablation, the median time between the preceding TACE and the ablation was 78 days (64 to 111 days, 25th percentile to 75th percentile).

### 3.3. All Treatment Delivered during Follow-Up

Table 3 outlines the most superior treatment modality received by each patient at any point of follow-up, irrespective of CR. The hierarchy of treatment was considered as follows based on conventional treatment algorithms: (1) liver transplant, (2) resection, (3) ablation, (4) SBRT, and (5) TACE. We considered SBRT as ‘superior’ to TACE in order to highlight the number of patients receiving this subsequent to initial TACE as well as some emerging evidence suggesting superior local control associated with SBRT compared to TACE [11,12,13].

Only a minority of TACE patients (42% in the overall cohort, 32% in the PSM cohort) never received a hierarchically superior treatment; however, this is still a remarkable proportion considering all patients at diagnosis had a solitary small tumour. The largest proportion of TACE patients received ablation as their ‘most superior’ treatment during the course of follow-up (44% in the overall cohort, 43% in PSM cohort), with further small numbers receiving transplant, SBRT, and resection.

In contrast, most ablation patients never received a hierarchically superior treatment (94% in the overall cohort, 91% in the PSM cohort), with only a few patients undergoing liver transplant or resection.

### 3.4. Outcomes

Outcomes in the original unmatched cohort are presented in Table 4 alongside the outcomes seen in the PSM cohort.

In the unmatched cohort, 13 out of 147 (9%) TACE patients underwent liver transplantation, with only 3 occurring as salvage treatment and the remaining 10 performed after later recurrence, while 5 out of 201 (2%) ablation patients required liver transplantation, with all of these required only after recurrent HCC. There was a total of 35 deaths, with 23 considered liver-related in the TACE group compared to 33, with 23 liver-related in the ablation group. OS and LRS were similar at the end of follow-up and at 1- and 3-year follow-up when comparing the two groups. There were significant differences seen in disease control when comparing the two groups with overall only 43 out of 147 (29%) of TACE patients both achieving and maintaining CR over follow-up compared to 95 out of 201 (47%) in the ablation group.

In the PSM cohort, 9 out of 115 (8%) TACE patients required liver transplantation, with only 2 occurring as salvage treatment and the remaining 7 performed after later recurrence, while 4 out of 115 (3%) ablation patients required liver transplantation, with all of these required only after recurrent HCC. There was a total of 26 deaths with 20 considered liver-related in the TACE group compared to 23 with 16 liver-related in the ablation group. OS and LRS were similar at the end of follow-up and at 1- and 3-year follow-up when comparing the two groups. However as with the unmatched cohort, there was a numerical trend towards improved disease control in the ablation group (41% achieving and maintaining CR vs. 32%). Further analysis is presented below.

### 3.5. Recurrence-Free Survival and Local Recurrence-Free Survival

In the propensity-score-matched cohort, recurrence-free survival and local recurrence-free survival were similar between the TACE and ablation groups (log rank test *p* = 0.355 and *p* = 0.390, respectively) when considered over the whole length of follow-up after their first CR, irrespective of when this was achieved. Recurrence-free survival curves are shown in Figure 2, and local recurrence-free survival curves in Figure 3. Median follow-up time was 20 months (597 days). There was some slight separation of the RFS survival curves seen between 12 and 24 months favouring ablation, with a non-significant trend towards improved RFS at 12 months (77.8% vs. 64.1%, log rank test *p* = 0.061). Differences in 1-year LRFS was less pronounced (77.8% vs. 66.1%, log rank test *p* = 0.110). In the original unmatched cohort, both RFS and LRFS were clearly superior in the ablation group (log rank test *p* = 0.001 and *p* = 0.002, respectively), with results presented in Appendix A.

### 3.6. Overall Survival

Overall survival was similar across the TACE and ablation groups (log rank test *p* = 0.652), with similar survival curves presented in Figure 4. Survival at the end of follow-up was 73.0% in the TACE group and 75.1% in the ablation group. Median overall follow-up time was 43 months (1308.5 days). There was similarly no significant difference in 1-year or 3-year overall survival (95.5% vs. 92.0%, *p* = 0.285; 82.2% vs. 80.9%, *p* = 0.713, respectively). Subgroup analyses involving Child–Pugh A and Child–Pugh B patients within the PSM cohort similarly demonstrated no significant differences between the two treatment groups (Appendix A). Sensitivity analysis was performed in the original unmatched cohort, with Kaplan–Meier survival analysis and log-rank test also failing to show a significant difference. These results are presented in Appendix A.

### 3.7. Liver-Related Survival

In the propensity-matched cohort, 16 out of 23 deaths (69.6%) in the ablation group were liver-related, compared to 20 out of 26 (76.9%) in the TACE group. Similar proportions were seen in the original unmatched cohort (23 out of 33 liver-related deaths (69.9%) in the ablation group, 27 out of 35 liver-related deaths (77.1%) in the resection group).

Figure 5 presents the results of Kaplan–Meier survival analysis. There was no significant difference in LRS between the two groups at the end of follow-up (81.6% vs. 77.7%, log rank test *p* = 0.495) or at 1- or 3-year follow-up (98.2% vs. 95.4%, *p* = 0.259; 87.9% vs. 84.8%, *p* = 0.539, respectively). There was similarly no difference in LRS in the original unmatched cohort (Appendix A).

### 3.8. All-Cause and Liver-Related Mortality with Liver Transplant as Competing Event

Competing-risks regression analysis was performed on the PSM cohort. Cumulative incidence function curves for all-cause mortality and liver-related mortality are presented in Figure 6 and Figure 7, respectively. After including liver transplant as a competing event, there was similar all-cause mortality (adjusted HR 0.87, 95% CI 0.50 to 1.52, *p* = 0.619) and liver-related mortality (adjusted HR 0.78, 95% CI 0.41 to 1.51, *p* = 0.469) when considering ablation compared to TACE. The results of the competing-risks regression analysis are presented in detail in Appendix A.

## 4. Discussion

In real-world clinical practice, the management approach in those with BCLC 0/A HCC with a small solitary tumour remains nuanced and individualised. While receipt of upfront curative therapy (namely, surgical resection, ablation, and liver transplantation) in such patients is recommended by treatment guidelines [14,15,16] and has recently been proposed as a quality indicator [17,18], many patients receive alternative treatments initially instead [19]. Despite the widespread use of TACE in those with small solitary tumour [4], the topic has received little attention in the literature other than a small number of studies that reported similar survival outcomes as compared with ablation [5,6,7,8,20]. In real-world practice, TACE is offered as an initial treatment with a discrete or implied plan for later future curative therapy such as with ablation. Our study is important in that it reports on a large multicentre cohort of Australian subjects and demonstrates that a treatment strategy involving initial TACE with subsequent treatment offered in the real-world environment results in similar survival outcomes to those receiving upfront ablation, calling into question the traditional dichotomy between TACE and ablation as non-curative and curative treatment modalities, respectively, and highlighting the role that initial TACE plays in real-world management.

While surgical resection is generally recommended as first-line treatment in BCLC 0/A patients where possible [14,15,16,21] and has been shown to be associated with superior survival outcomes compared to non-surgical treatment [22], many patients are not suitable for resection due to portal hypertension, unsuitable anatomy, severity of liver disease, or non-liver comorbidities. Similarly, ablation can be considered undesirable due to tumour location such as an inaccessibly high sub-diaphragmatic location, proximity to major blood vessels or bile ducts, and proximity to other intra-abdominal organs. Additionally, in centres where ablation is performed only under ultrasound guidance, sonographic visibility of the tumour affects perceived suitability for the procedure. Furthermore, patient and clinician preferences for or against certain treatments, differential consideration of risks associated with different treatments, and also centre-dependent logistics (such as availability or time to treatment for different modalities) play a key role in determining treatment selection. While liver transplant is recommended by some international guidelines [14,16] as the preferred alternative curative modality, its real-world use is limited by donor organ availability and in Australia has generally been reserved for patients with multinodular disease, recurrent HCC, decompensated cirrhosis, or as a salvage treatment. TACE is therefore often used in patients with single solitary tumour who are not suitable for resection or ablation, or otherwise require upfront transplantation. We performed our real-world propensity-score-matched study in order to understand if survival differed between those undergoing upfront ablation and those receiving initial TACE.

As expected, we found systematic differences in the ablation- and TACE-treated groups. Although only patients with solitary tumours 3 cm or less were included, there was still a statistically significant difference in tumour size, with patients receiving TACE having larger tumours on average. Accordingly, 41% of TACE patients had BCLC-0 HCC as compared with 52% of ablation patients. This is to be expected, as generally speaking, smaller tumour size favours treatment allocation to ablation over TACE; however, it should be noted that all single tumours 3 cm or less are considered amenable to ablation on the basis of size alone [23]. The only other significant difference demonstrated was in liver disease aetiology, with upfront ablation patients more likely to have alcohol as a cause. Because patients with alcohol-related liver disease may be perceived as at greater risk for loss to follow-up, an upfront curative treatment may be preferred to a treatment strategy that relies on a non-curative treatment with explicit or implicit plans for future further treatment. It should also be mentioned that we have observed that liver transplant centres in Australia are more likely to treat BCLC 0/A HCC with TACE initially [4] and are also less likely to have patients with alcohol-related liver disease, which may be the primary driver for this difference. Age, sex, diabetes, smoking, platelet count, liver disease severity, and CCI were all otherwise statistically similar between the two groups. Data collection regarding complications was limited across the two groups, but no deaths or major complications were recorded.

One in five patients who received initial TACE never achieved CR compared to approximately one in seven of those who received initial ablation. Of those TACE patients who did achieve CR, most did so after one or more TACE alone, but a significant number required follow-up ablation. Even in those who did achieve CR directly from TACE, a large proportion of these patients went on to receive ablation later, often for treatment of local recurrence. TACE prior to ablation has been shown to improve post-ablation outcomes [24,25], with multiple reasons postulated for this phenomenon. Firstly, TACE targets a broader area of liver than ablation therapy and is therefore likely to target viable micrometastases adjacent to the tumour that would otherwise be outside of the ablation zone and hence go untreated. Secondly, TACE disrupts hepatic arterial flow which reduces the potential heat sink effect at the time of ablation, increasing the size of the ablation zone and thereby increasing the likelihood of complete ablation of the tumour. Lastly, the intra-tumoral deposition of lipiodol at time of TACE can subsequently improve tumour visualisation at the time of ablation. It should be noted that most of the evidence for combination TACE and ablation has been developed within a time-interval of 2 weeks; however, in our cohort, there was a median delay between the treatments of 78 days. The impact of time between TACE and ablation warrants further investigation.

Before performing our survival analysis, we utilised propensity score matching, a quasi-experimental technique to maximise comparability between the two groups and thereby minimise confounding, in order to produce a total cohort of 230 patients, or 115 matched pairs. After matching, there were no significant differences between the two groups. We then performed Kaplan–Meier survival analysis with log-rank test in the propensity-matched cohort assessing for differences in overall survival, liver-related survival, and recurrence-free survival. We found that patients undergoing initial TACE had similar overall survival, liver-related survival, recurrence-free survival, and local recurrence-free survival when compared to those receiving upfront ablation. After performing competing-risks regression using liver transplant as a competing event, the groups’ risk of death of any cause and liver-related death remained similar, with no significant differences seen, affirming that the similarity in survival outcomes was not an artefact relating to systematic differences in liver transplant utilisation between the two groups. This is a remarkable finding, as it provides compelling evidence that a treatment approach employing initial TACE has a role in even those with small solitary tumour and confers similar survival benefit to upfront ablation when allowing for tailored subsequent treatment including with follow-up ablation.

Of note, unadjusted analysis performed in the unmatched cohort did show that recurrence-free survival and local recurrence-free survival was inferior in patients not treated with upfront ablation, but even this did not translate to differences in overall survival, potentially highlighting that any failure in local disease control can be made up for with subsequent appropriate follow-up treatment. With the evidence for reasonable medium- to long-term survival associated with initial TACE provided by our study, as well as increasingly strong evidence for other novel modalities such as SBRT [26,27,28], we believe that further work is needed to prospectively compare the current existing treatment strategies with the results used to inform future treatment guidelines and algorithms.

While our study is strengthened by its propensity-score-matched design, our study does have major limitations. Firstly, our study is retrospective, which inherently carries a risk of information bias, in particular relating to the documentation of tumour burden and response assessments, with the reliability of 1 month documentation of CR potentially limited by the presence of lipiodol or post-treatment inflammatory change. Secondly, due to the limitations of the data capture, clear assessment of tumour location and the implications on suitability for ablation was not able to be assessed, which may introduce a component of unaddressed selection bias. Furthermore, the intention and reasons behind selection of initial TACE treatment, such as whether or not it was explicitly planned as a bridge or downstaging therapy prior to ablation or transplant or other treatment, has not been collected and therefore could confound our results. In a similar vein, the specific reason why upfront ablation was not used, such as tumour location, visibility on ultrasound, or presence of perihepatic ascites, has also not been clearly elucidated in the data collection. Lastly, our study was limited in patient numbers and follow-up time. While we had a total of 230 patients available for analysis in the PSM cohort and a median follow-up time of 43 months, it is possible that with a greater number of patients or longer follow-up time, we may have seen a significant difference in survival develop between the ablation and TACE groups, particularly in RFS and LRFS, although we believe that this is less likely based on the trends seen in our analysis.

## 5. Conclusions

In a real-world cohort of Australian patients with small solitary HCC, we found that patients undergoing TACE rather than ablation as initial treatment had similar survival outcomes, even when considering liver transplant as a competing risk. Our study provides valuable evidence for the utility of initial TACE as a treatment option in BCLC 0/A HCC. Prospective studies are needed to provide more rigorous evidence regarding the comparative efficacy and safety of the two treatment approaches we have described in this retrospective cohort and verify our finding that survival outcomes are similar. Further work is needed to better define the optimal use of TACE in this cohort, including the role for combination therapy with ablation, including patient selection and timing.

## Figures and Tables

**Figure 1 cancers-16-03010-f001:**
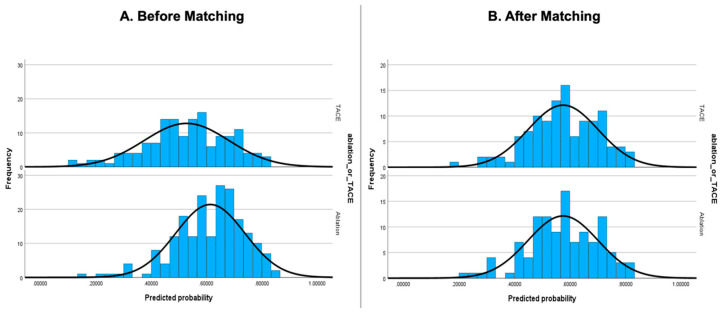
Distribution of propensity scores in ablation and TACE groups before and after propensity score matching.

**Figure 2 cancers-16-03010-f002:**
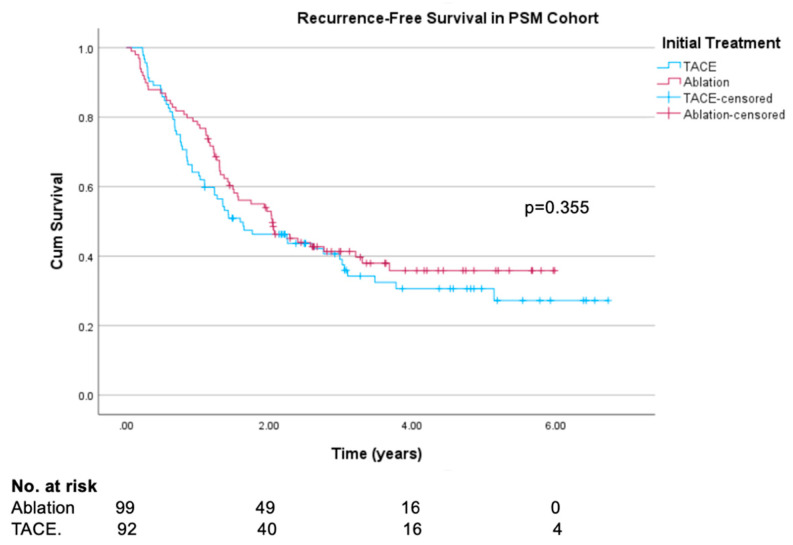
Kaplan–Meier recurrence-free survival curves in propensity-score-matched ablation and TACE groups.

**Figure 3 cancers-16-03010-f003:**
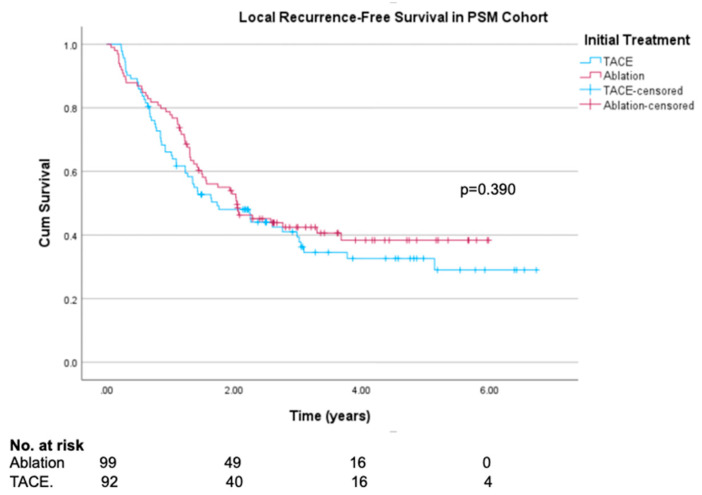
Kaplan–Meier local recurrence-free survival curves in propensity-score-matched ablation and TACE groups.

**Figure 4 cancers-16-03010-f004:**
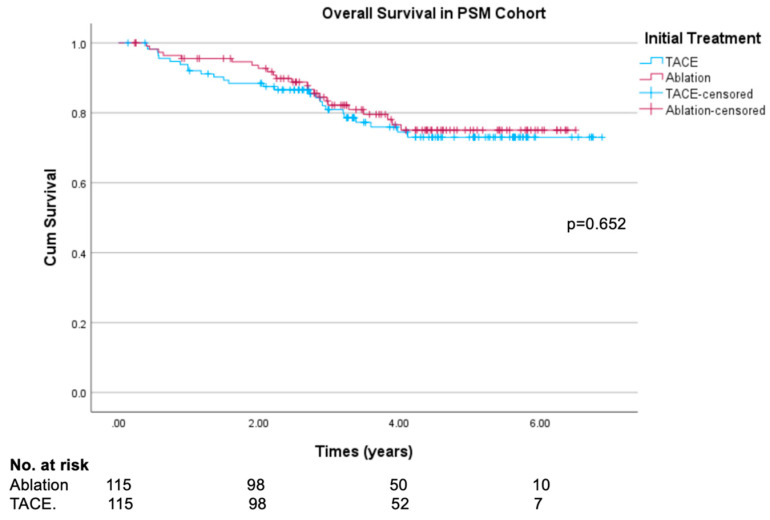
Kaplan–Meier overall survival curves in propensity-score-matched ablation and TACE groups.

**Figure 5 cancers-16-03010-f005:**
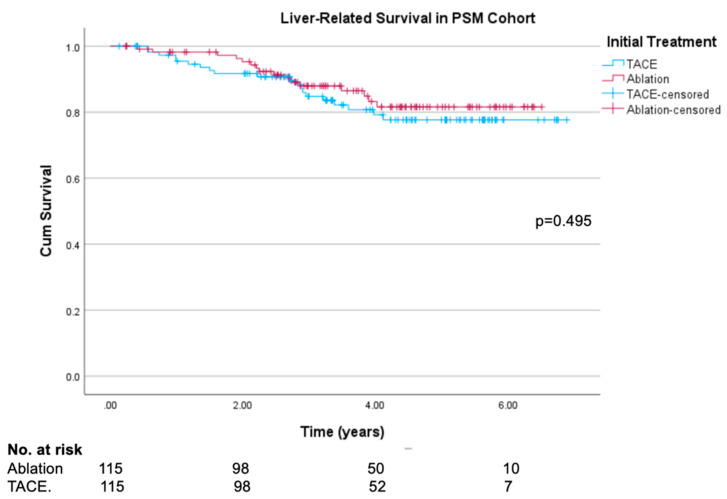
Kaplan–Meier liver-related survival curves in propensity-score-matched ablation and TACE groups.

**Figure 6 cancers-16-03010-f006:**
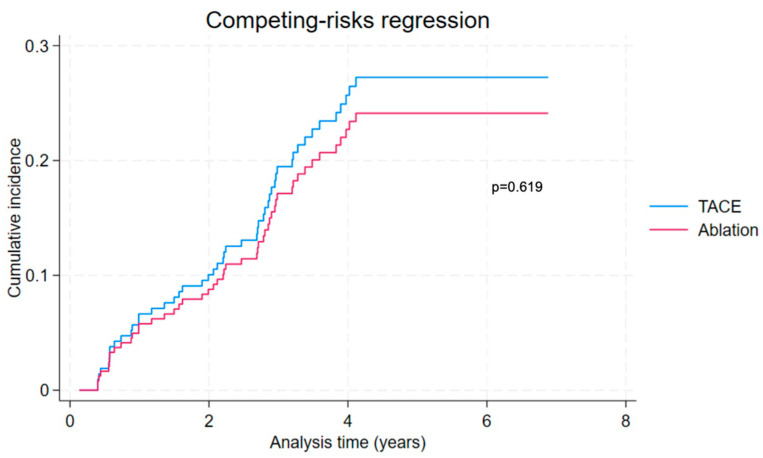
Competing-risks regression considering all-cause mortality with transplant as competing risk in propensity-score-matched ablation and TACE groups.

**Figure 7 cancers-16-03010-f007:**
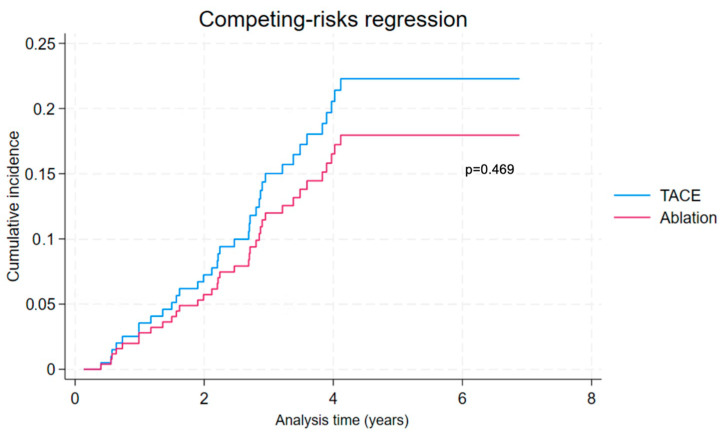
Competing-risks regression considering liver-related mortality with transplant as competing risk in propensity-score-matched ablation and TACE groups.

**Table 1 cancers-16-03010-t001:** Patient characteristics before and after propensity score matching.

	Matched (*n* = 230)			Unmatched (*n* = 348)		
Characteristic	Ablation *n* = 115	TACE *n* = 115	*p*-Value	Ablationn = 201	TACE *n* = 147	*p*-Value
Age *	62.6 ± 11.7	63.8 ± 9.7	0.409	64.3 ± 10.8	62.9 ± 12.05	0.255
Sex			0.337			0.443
Female	22 (19%)	28 (24%)		48 (24%)	30 (20%)	
Aetiology			0.201			0.005
Alcohol	17 (15%)	11 (10%)		38 (19%)	12 (8%)	
HBV	13 (11%)	9 (8%)		18 (9%)	12 (8%)	
HCV	22 (19%)	24 (21%)		35 (17%)	29 (20%)	
MASLD	14 (12%)	10 (9%)		26 (13%)	11 (8%)	
Other	3 (3%)	9 (8%)		5 (3%)	13 (9%)	
metALD	7 (6%)	9 (8%)		10 (5%)	11 (8%)	
HBV/HCV	5 (4%)	3 (3%)		8 (4%)	3 (2%)	
HCV + SLD	33 (29%)	33 (29%)		55 (27%)	45 (31%)	
HBV + SLD	1 (1%)	7 (6%)		6 (3%)	11 (8%)	
Diabetes			0.657			0.575
Yes	30 (26%)	33 (29%)		56 (28%)	137 (25%)	
Smoking			1.000			0.396
Yes	36 (31%)	36 (31%)		66 (33%)	42 (29%)	
Tumour size (cm) **	2.0 (1.5 to 2.5)	1.9 (1.6 to 2.4)	0.875	1.8 (1.4 to 2.3)	2.1 (1.6 to 2.6)	<0.001
Platelet (×10^9^ /L) **	105 (75 to 155)	112 (77 to 162)	0.621	113 (80 to 155)	107 (70 to 156)	0.272
CCI **	5 (3 to 6)	5 (4 to 6)	0.444	5 (3 to 6)	5 (4 to 6)	0.329
Child–Pugh Score			0.991			0.078
5	63 (55%)	63 (55%)		103 (51%)	74 (50%)	
6	31 (27%)	31 (27%)		62 (31%)	38 (26%)	
7	12 (10%)	12 (10%)		24 (12%)	14 (10%)	
8	7 (6%)	6 (5%)		9 (5%)	12 (8%)	
9	2 (2%)	3 (3%)		3 (2%)	9 (6%)	
BCLC			0.510			0.059
0	54 (47%)	59 (51%)		104 (52%)	61 (41%)	
Specific modality						-
MWA	109 (95%)	-		162 (81%)	-	
PEI	0	-		2 (1%)	-	
RFA	6 (5%)	-		37 (18%)	-	
DEBTACE	-	11 (10%)		-	16 (11%)	
cTACE	-	104 (90%)		-	131 (89%)	

HBV, hepatitis B virus; HCV, hepatitis C virus; MASLD, metabolic-dysfunction-associated steatotic liver disease; metALD, metabolic and alcohol-related liver disease; CCI, Charlson Comorbidity Index; BCLC, Barcelona Clinic Liver Cancer; RFA, radiofrequency ablation; MWA, microwave ablation; TACE, trans-arterial chemo-embolisation. * mean ± standard deviation. ** median (25th percentile—75th percentile).

**Table 2 cancers-16-03010-t002:** Sequential treatments prior to documented CR in the propensity-score-matched and original unmatched cohort.

	Matched (*n* = 230)		Unmatched (*n* = 348)	
CR Achieved	Ablation *n* = 115	TACE *n* = 115	Ablation *n* = 201	TACE *n* = 147
Never After initial treatment After 2 or more treatments Multiple ablation Multiple TACE Combination of TACE and ablation Combination with other locoregional * With eventual resection With eventual liver transplant	16 (14%) 85 (74%) 6 (5%) - 5 (4%) 1 (1%) 2 (2%) 0	23 (20%) 64 (56%) - 11 (10%) 14 (12%) 0 1 (1%) 2 (2%)	29 (14%) 151 (75%) 9 (5%) - 8 (4%) 2 (1%) 2 (1%) 0	30 (20%) 78 (53%) - 13 (9%) 22 (15%) 0 1 (1%) 3 (2%)

TACE, trans-arterial chemo-embolisation. * stereotactic body radiation therapy (SBRT) and selective internal radiation therapy (SIRT)

**Table 3 cancers-16-03010-t003:** Most superior treatment modality delivered at any point during follow-up in the propensity-score-matched and original unmatched cohort.

	Matched (*n* = 230)		Unmatched (*n* = 348)	
	Ablation *n* = 115	TACE *n* = 115	Ablation *n* = 201	TACE *n* = 147
Liver transplant Resection Ablation SBRT TACE	4 (3%) 6 (5%) 105 (91%) - -	9 (8%) 2 (2%) 50 (43%) 6 (5%) 48 (32%)	5 (2%) 7 (3%) 189 (94%) - -	13 (9%) 2 (1%) 64 (44%) 6 (4%) 62 (42%)

**Table 4 cancers-16-03010-t004:** Summary of overall, 1-year, and 3- year outcomes in the PSM and original unmatched cohort.

	Matched Cohort (*n* = 230)			Unmatched Cohort (*n* = 348)		
Outcomes	Ablation *n* = 115	TACE *n* = 115	*p*-Value	Ablation *n* = 201	TACE *n* = 147	*p*-Value
CR never achieved	16 (14%)	23 (20%)	0.219	29 (14%)	30 (20%)	0.142
Recurrence in those who achieved CR						
Never	47 (47%)	37 (40%)		95 (55%)	43 (37%)	
Local recurrence	50 (51%)	41 (55%)		71 (41%)	68 (58%)	
Distant recurrence	2 (2%)	4 (4%)		6 (4%)	6 (5%)	
Total number of deaths	23 (20%)	26 (23%)		33 (16%)	35 (24%)	
Liver-related	16 (14%)	20 (17%)		23 (11%)	27 (18%)	
Non-liver-related	7 (6%)	6 (5%)		10 (5%)	8 (5%)	
Transplant	4 (3%)	9 (8%)		5 (2%)	13 (9%)	
Before CR	0	2 (2%)		0	3 (2%)	
After CR and before recurrence	0	0		0	0	
After recurrence	4 (3%)	7 (6%)		5 (2%)	10 (7%)	
Overall survival						
At 1 year	95.5%	92.0%	0.285	95.8%	91.0%	0.073
At 3 years	82.2%	80.9%	0.713	85.1%	80.2%	0.200
At the end of follow-up	75.1%	73.0%	0.652	79.1%	72.1%	0.129
Liver-related survival						
At 1 year	98.2%	95.4%	0.259	97.8%	95.0%	0.158
At 3 years	87.9%	84.8%	0.539	89.5%	84.4%	0.192
At the end of follow-up	81.6%	77.7%	0.495	84.8%	76.8%	0.095
Recurrence-free survival						
At 1 year	77.8%	64.1%	0.061	79.7%	62.4%	<0.001
At 3 years	41.3%	39.1%	0.519	51.8%	37.7%	0.001
At the end of follow-up	35.8%	27.2%	0.355	44.8%	24.6%	0.001
Local recurrence-free survival						
At 1 year	77.8%	66.1%	0.110	80.8%	64.0%	0.002
At 3 years	42.5%	39.4%	0.561	53.8%	38.0%	0.004
At the end of follow-up	38.3%	29.0%	0.390	48.7%	28.9%	0.002

CR, complete response.

## Data Availability

The data presented in this study are available on request from the corresponding author. The data are not publicly available due to privacy concerns.

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
