# Peer review of "Initial Trans-Arterial Chemo-Embolisation (TACE) Is Associated with Similar Survival Outcomes as Compared to Upfront Percutaneous Ablation Allowing for Follow-Up Treatment in Those with Single Hepatocellular Carcinoma (HCC) ≤ 3 cm: Results of a Real-World Propensity-Matched Multi-Centre Australian Cohort Study"

_cancers, 2024, doi:10.3390/cancers16173010_

Round 1

Reviewer 1 Report

Comments and Suggestions for Authors

I appreciate for giving me a chance to review your valuable manuscript. The manuscript discusses that patients undergoing TACE rather than ablation as initial treatment had similar survival outcomes, even when considering liver transplant as a competing risk in the retrospective cohort. These findings contribute to the understanding of optimal treatment approaches for early stage HCC patients receiving TACE.

I have some comments to improve your research. Please check and reconsider those. Thank you.

1.  In the introduction part, some studies already detect that TACE showed similar outcomes as compared to ablation. Compared with previous studies, what is original in your manuscript and what aspect could you clarified? Please reconsider and modify it.

2.  In the method part, there appears to be no description or citation for the Charlson Comorbidity Index (CCI). Please confirm this point.

3.  In the discussion, the conclusions of this study are unclear. Based on the results of the present study, should TACE be aggressively performed because it does not worsen the prognosis, or should it not be performed because there is no difference in local recurrence rates with TACE? What is your opinion?

Minor amendments: Extra period in line 193. 

Author Response

Reviewer 1

1) In the introduction part, some studies already detect that TACE showed similar outcomes as compared to ablation. Compared with previous studies, what is original in your manuscript and what aspect could you clarified? Please reconsider and modify it.

In order to highlight the limitation of prior literature and the novelty of our study, we have changed the introduction to read “Only a small number of studies(5-8) have reported on this question, with evidence of similar outcomes as compared to ablation, however the generalizability of these findings requires further verification, as all of these studies were single-centre and none of them reported on subsequent follow-up treatment. In order to describe the real-world use and outcomes associated with a treatment strategy involving initial TACE for single small HCC, together with and without further follow-up treatment, and provide more robust evidence in comparing its impact on survival outcomes as with upfront ablation, we performed this multi-centre retrospective observational propensity-score matched cohort study”

2) In the method part, there appears to be no description or citation for the Charlson Comorbidity Index (CCI). Please confirm this point.

We have added the reference to the first mention of CCI. (Charlson ME, Pompei P, Ales KL, MacKenzie CR. A new method of classifying prognostic comorbidity in longitudinal studies: development and validation. J Chronic Dis. 1987;40(5):373-83)

3) In the discussion, the conclusions of this study are unclear. Based on the results of the present study, should TACE be aggressively performed because it does not worsen the prognosis, or should it not be performed because there is no difference in local recurrence rates with TACE? What is your opinion?

We stand by the stated conclusions of our study as follows: “Our study provides valuable evidence for the utility of initial TACE as a treatment option in BCLC 0/A HCC. Prospective studies are needed to provide more rigorous evidence regarding the comparative efficacy and safety of the two treatment approaches we have described in this retrospective cohort and verify our finding that survival outcomes are similar. Further work is needed to better define the optimal use of TACE in this cohort, including the role for combination therapy with ablation, including patient selection and timing.” We believe the main take away of our study is validating that there is a role for TACE in single small HCC (rarely acknowledged by the guidelines) usually when upfront curative treatment is not feasible, but we believe further work is needed to better define the role for TACE in these patients.

Reviewer 2 Report

Comments and Suggestions for Authors

The authors describe “Initial trans-arterial chemo-embolization (TACE) is associated with similar survival outcomes as compared to upfront percutaneous ablation allowing for follow-up treatment in those with single hepatocellular carcinoma (HCC) ≤ 3cm: results of a real-world”  The title is impressive, and these study results have potential usefulness in future medicine. However, some concerns should be addressed.

Major Points

In the first point, this study included cases of poor liver function up to a Child-Pugh Score of 8.

In this population, the degree of deterioration of liver function after TACE and local therapy varies according to liver reserve capacity.

In addition, this study was conducted in a population with a small tumor volume, so the prognostic impact of poor hepatic reserve is extremely strong.

Therefore, the authors should examine the prognosis of the three groups of Child-Pugh A5, A6, or more by the liver reserve.

Author Response

Reviewer 2

1) In the first point, this study included cases of poor liver function up to a Child-Pugh Score of 8. In this population, the degree of deterioration of liver function after TACE and local therapy varies according to liver reserve capacity. In addition, this study was conducted in a population with a small tumor volume, so the prognostic impact of poor hepatic reserve is extremely strong. Therefore, the authors should examine the prognosis of the three groups of Child-Pugh A5, A6, or more by the liver reserve.

We believe that we have fully addressed this by using Child-Pugh Score as a key variable in the propensity-score matching process. In the propensity-score matched cohort, the Child Pugh Score is very well balanced between the two groups (see Table 1, proportions near identical, p value of 0.991) and by matching the two groups so closely, we have effectively adjusted for the differential impact of CPS on mortality.

Round 2

Reviewer 2 Report

Comments and Suggestions for Authors

Selection bias due to tumor localization naturally arises when ablation and TACE are performed, and the author should consider that the content of the study should include tumor localization.

However, the most problematic aspect of this study is that it includes patients with poor hepatic reserve capacity regardless of whether or not propensity matching was performed.

The author must perform stratifying prognosis by reserve capacity is essential.

Author Response

1) Selection bias due to tumor localization naturally arises when ablation and TACE are performed, and the author should consider that the content of the study should include tumor localization. 

We agree with the Reviewer that this is an important limitation of our study and have highlighted this in the last paragraph of the discussion as follows: “Secondly, due to the limitations of the data capture, clear assessment of tumour location and the implications on suitability for ablation was not able to be assessed which may introduce a component of unaddressed selection bias.”

 2) However, the most problematic aspect of this study is that it includes patients with poor hepatic reserve capacity regardless of whether or not propensity matching was performed. The author must perform stratifying prognosis by reserve capacity is essential.

We completely agree with the Reviewer of the importance of hepatic reserve and have presented stratified analysis of overall survival according to Child Pugh Class in the supplementary material (Figures S3 and S4) and reproduced the same result of no significant survival difference in both the Child-Pugh A and Child-Pugh B subsets of the PSM cohort. We maintain that propensity score matching using Child Pugh score is a rigorous and appropriate method of minimizing the potential confounding effect of diminished hepatic reserve on survival (Austin PC. ‘An Introduction to Propensity Score Methods for Reducing the Effects of Confounding in Observational Studies’. Multivariate Behav Res. 2011 May;46(3):399-424. doi: 10.1080/00273171).

Round 3

Reviewer 2 Report

Comments and Suggestions for Authors

The authors answered all of the concerns raised by the reviewer. I have no comment.